# Forgiveness in Committed Couples: Its Synergy with Humility, Justice, and Reconciliation

**Everett L. Worthington Jr. [1,\*], Eric M. Brown [2] and John M. McConnell [2]** 

1    Department of Psychology, Virginia Commonwealth University, Richmond, VA 23284, USA
2    School of Psychology, Counseling and Family Therapy, Wheaton College, Wheaton, IL 60187, USA;
     eric.brown@wheaton.edu (E.M.B.); john.mcconnell@wheaton.edu (J.M.M.)
\*    Correspondence: eworth@vcu.edu; Tel.: +1-804-828-1150

**Abstract:** Theologians, pastors, and psychological help-providers have not always worked harmoniously. This can be especially true with couples. Theological and pastoral help-providers value marriage as sacred and are reluctant to entertain ending it. Most psychotherapists have more training and experience in individual psychotherapy than in couple therapy. Drawing on the parable of the Good Samaritan, we appeal to theologians, pastors, and psychological help-givers to work together. We examine ways that psychological findings might inform theology and pastoral practice. As an example, we use forgiveness in committed romantic relationships. What causes strong couple relationships are the formation, strengthening, maintenance, and (when damaged) repair of ruptures in the emotional bond. Thus, forgiveness is one major cause of good marriage. Forgiveness requires being oriented toward the other person's welfare, and in humility responding to wrongdoing mercifully. Forgiving in committed relationships seeks a net positive emotional valence toward the partner built on empathy, humility, and responsibility. Good relationships also involve self-forgiveness when one feels self-condemnation over one's own misdeeds. For help-givers, humility is a key to promoting relational experiences of virtue. We show that forgiveness is related to health. Religiously oriented help-providers can promote better relationships and better health by fostering forgiveness.

**Keywords:** couples; religion; treatment; forgiveness; humility; collaboration; empathy

## 1. Introduction

> He has told you, O man, what is good;
> and what does the LORD require of you but to do justice, and to love kindness,
> and to walk humbly with your God?
> (Micah 6:8, All Scriptures in this article are in the English Standard Version; ESV)

In a talk by Jakes (2011), he used the parable of The Good Samaritan (Luke 10:30–35) to discuss interdisciplinary collaboration. The relationship between the Samaritan and the innkeeper presents an interesting parallel between pastors and counselors. The Samaritan, who is not a priest, is really boots-on-the-ground "doing church." He helps the needy person, yet recognizes his own limitations—he cannot care for the wounded man alone. He takes the person to a helper with different gifts and skills, and leaves him there for longer-term care and healing. Although Jakes paralleled the pastor to the Samaritan, doing emotional and spiritual first-aid, and the innkeeper to the counselor, doing long-term psychological care, the collaborative relationship could easily occur in the other direction. Different helping professions can work together. Pastors and counselors each have different skill sets, and yet together, the professions help the person of faith. They also can inform one another

about theological and psychological principles to improve their individual and collaborative efforts. We believe three general professions—theology, pastoral care and counseling, and psychological help-giving (e.g., clinical and counseling psychologists, marital and family therapists, mental health counselors, and clinical social workers) can work together most harmoniously when they understand each other's core knowledge base and technical approaches. As psychological help-givers, we offer theology and pastoral care some foundational principles and approaches of forgiveness in committed relationships in an effort to better bridge the gap between our professional silos.

The fact is, though, that these professions do not always collaborate well with each other. Weaver et al. (1997) found that out of 2468 quantitative studies published in APA journals, only four included clergy in their data. Since 1997, a burgeoning interest in the collaboration of mental health professionals and clergy has emerged (Oxhandler and Parrish 2018; Oxhandler et al. 2018). Collaboration between clergy and psychotherapists may take several forms, the most common being that of clergy consulting with psychotherapists on how to develop a peer counseling ministry or establish conflict-resolution guidelines within their congregation (McMinn et al. 2003). Another common form of partnership entails licensed mental health therapists consulting with clergy when providing workshops for congregations on topics such as marriage enrichment, parenting skills, and personal growth (McMinn et al. 2003; Breuninger et al. 2014).

Some therapists build relationships with local clergy so they can consult on cases where the client is religious. This may entail a more complex form of collaboration when the therapist and the client's clergyperson are working together on a clinical case. Rarely utilized, a psychologist may consult for clergy wishing to conduct a needs assessment of their church context, develop psychologically and theologically integrated programs, and empirically evaluate ministry programs. Psychotherapists and clergy report several components that are critical to positive collaborations: (a) trust between them; (b) mutual respect training and expertise; (c) a shared goal focused on the client's therapeutic healing; (d) ongoing informed consent from the client, which ideally should include a signed release form from the client for clergy and therapist communication (Barnett and Johnson 2011; McMinn et al. 2003). Ethical concerns center on appropriate informed consent, how to document religious and spiritual services for insurance reimbursement, and boundary issues between differing professions (Barnett and Johnson 2011). There are several reasons why theologians, clergy, and psychological help-givers do not necessarily work well together. First, pastors typically feel responsible for the spiritual status of their parishioners, and surveys have found that psychologists and other mental health workers are not personally religious on average (e.g., Bergin and Jensen 1990). Meylink and Gorsuch (1988) reviewed research on the relationships between psychotherapists and clergy. About 40 percent of people who seek help first approach clergy. Yet less than 10 percent of those help-seekers were referred to mental health professionals. On the other hand, psychologists almost never referred clients to clergy or other religious resources. Recently, more openness to two-way referral flow has been found (McMinn and Campbell 2008). Recent models promoting collaboration between clergy and psychologists have drawn on the belief that for clergy and psychotherapists to work well together, they must draw on shared religious values (Pargament 2007; Plante 2009). Thus, pastors might be reluctant to refer because they worry that the spiritual status of the parishioners might be undermined by a spiritually dissimilar mental health provider. Research has not shown this to be an often-substantiated worry (Oxhandler and Parrish 2018), but that does not stop pastors from being concerned. Second, pastors might believe they are competent to provide the counsel that a parishioner needs. Research has shown that pastors receive little formal training in psychotherapy, counseling, or couple therapy over the course of their training (Payne 2009). Although some competence can be developed by sheer breadth of experience, research in counselor training has found mere experience often makes people more confident in their counseling abilities and yet does not change their actual competence much (Goldberg et al. 2016). Rather, it is supervised experience in counseling that accelerates counseling competency more than mere experience. Third, similarly, mental health professionals often need

training in how to deal with spiritual issues (McMinn and Campbell 2008; Worthington et al. 2009). Fourth, psychotherapists—with the exception of couple and family therapists—typically do not receive much training in couple therapy, even though many cases they see will likely involve marital issues at some point. Fifth, psychologists and other mental health professionals might not trust pastors to give good mental health counseling and thus are often reluctant to refer, even when there is a religious, theological, or spiritual problem that the mental health provider is not competent to deal with (Breuninger et al. 2014). Sixth, theologians rarely counsel, yet they provide consultation on religious and spiritual matters for both pastors and mental health providers. For the most part, theologians—Biblical theologians, systematic theologians, and other types of theologians except for practical theologians—do not usually deal with emotional and relational problems daily. They might consider such topics, but they often do so from the standpoint of providing logically consistent interpretations of Scripture and other religious interpretations. Seventh, pastors and theologians often assume that psychotherapists are more willing to counsel couples to divorce because they might not value marriage as a sacred covenant. In Western culture, people increasingly treat marriage as a contract rather than a sacred covenant, yet churches tend to be among the more conservative institutions (Ripley et al. 2005).

We suggest theologians, clergy, and psychological help-givers can collaborate more often to not only improve care of parishioners but also increase each other's theological and counseling competencies through consultation. Better interdisciplinary collaboration and rapport building will help dispel myths, and overall lead to more effective interdisciplinary helping models that are spiritually and psychologically integrated. Often, such collaboration can be initiated by a mental health professional's offer to assess the congregation (Dominguez and McMinn 2003). Long-term stable couple relationships, whether marriage or civil union, are frequently under strain. Couples marry or enter those unions believing they will not divorce, desiring to live together permanently, and striving to create lasting unions. Yet, external and internal pressures often mitigate against such long-term stability. We encourage the professionals in these diverse fields to work together collaboratively, refer to each other more often, consult with and educate each other, and further develop helping models together. We believe that such collaborations can save many troubled relationships.

We seek to provide this encouragement to collaborate not just by exhortation. In this article, we focus specifically on improving this collaboration to better promote the spiritual and emotional care of couples in conflict. We provide critical information about couple problems and couple therapy within the framework of developing virtues—often called spiritual formation within religious and spiritual circles. To these ends, we have three objectives. By the end of the article, we hope the reader will be able to (1) know what psychological research says about the cause of good marriages or committed relationships, (2) describe what forgiveness is, particularly in committed romantic relationships, and (3) use evidence-based therapeutic practices to promote forgiveness and self-forgiveness in their pastoral or psychotherapeutic care: a five-step method to REACH Forgiveness (Worthington 2006) and a six-step method to forgive oneself responsibly (Griffin et al. 2015a). We discuss how humility, justice, and reconciliation play important roles in the process of forgiveness in committed romantic relationships.

## 2. Results

### 2.1. What Causes a Good Marriage or Long-Term Committed Romantic Relationship?

The state of the clinical science regarding marriage and long-germ relationships in 1993 could be summarized simply. The field was enamored with the Gottman ratio (Gottman 1993). Gottman advanced a radical idea. He claimed that if one examines as little as 10 min of videotape of a couple interacting and tallies the number of positive interactions relative to the number of negative interactions, a 5 to 1 ratio of positive to negative would predict a happy, stable couple with 94 percent success. In addition, a ratio lower than 5:1 tends to be dramatically lower—a step function to the level of 2:1, 1:1, or lower. Many clinicians, pastors, and theologians drew a couple of conclusions. First, they understood

Gottman to be claiming that the cause of a poor marriage is too low a ratio of positive-to-negative interactions during communication and conflict management. Second, some couple therapists, pastoral counselors, and theologians assumed that the way to make a better relationship was to increase the ratio. If a couple had 10 positives and four negatives (2.5 to 1 ratio), then the couple could become a happy couple if they increased their positives to 20 (resulting in a 20 to 4 ratio, i.e., 5:1, or decreased their negatives to 2 (resulting in a 10 to 2 ratio, i.e., 5:1). Obviously, it is easier to reduce the negatives by two than to increase the positives by 10. So, some people who helped couples concluded that helpers should train couples in more positive communication strategies, and especially in conflict resolution strategies (i.e., reducing negatives).

By the early 2000s, about 10 years of research had revealed these to be mistaken conclusions (Gurman and Fraenkl 2002). We know now that the 5:1 ratio is not the *cause* of a good marriage. It is the *effect* of a good marriage. It is not so much skills, communication, or conflict resolution per se that makes a relationship better. It is the ability to control and limit any negative emotional climate and restore a positive emotional climate as needed. Good marriages are caused when couples can form, maintain, strengthen, and repair their emotional bonds. When couples do that, they will have better communication and resolve conflict better, resulting in a stronger positive to negative ratio.

## 2.2. Hope-Focused Couple Approach to Enrichment and Counseling

Worthington and his colleagues have developed and investigated the Hope-Focused Couple Approach (Ripley and Worthington 2014), which takes seriously findings by Gottman and those researchers who showed that emotional bonds, attachments from childhood into adulthood, and emotional experience and expression are the keys to causing better committed couple relationships. The full Hope-Focused Couple Approach, which is not our main concern here, has been described in detail by Worthington (2005) and by Ripley and Worthington (2014). It argues that hope is necessary for a good relationship. According to Snyder (1994), hope is seen as the sum of willpower (i.e., agency to change), plus waypower (i.e., knowledge of methods to bring change). However, as we know from theologian Gabriel Marcel (Marcel 1962; see also Bury et al. 2016; Crabb 2001), hope also involves expectant waiting even when we do not see change occurring. In addition, Worthington teaches not just strategies and methods that are reactive to couples' problems, but strategies and methods that use a proactive approach: to promote faith, work, and love. They divide the approach into two halves (Worthington 2005). First, the approach teaches Handling Our Problems Effectively (HOPE) (Worthington 2006), which helps couples use the time-honored methods of improving communication, conflict resolution, and intimacy-building. In the second half, Forgiveness and Reconciliation through Experiencing Empathy (FREE), the approach helps couples try to recover from breaches in their trust and the unforgiveness that often arises (Ripley et al. 2014; Worthington et al. 2015). Due to space limitations, we will say more about forgiveness, however, than about reconciling.

## 2.3. Forgiveness

### 2.3.1. Injustice Gap

When people are hurt or offended, they almost automatically begin to make unconscious cognitive computations. They measure their sense of an injustice. The *injustice gap* is defined as the difference between the way we would like to have an offense resolved and the way we perceive the status at the present (Exline et al. 2003). The size of injustice gap is proportional to the difficulty in forgiving. Thus, big hurts or big offenses are difficult to forgive, and small hurts or offenses are easy to forgive. People keep this computation current as other offense-related events unfold. For example, suppose a husband has an injustice gap of say (arbitrarily) 50 units of perceived injustice due to a romantic betrayal. The husband asks his wife why she did such a horrid act. She responds, "Because I hate you." The injustice gap will be widened far beyond what it was originally. But if, when the husband queries his wife, she instead cries pitifully, apologizes sincerely, begs for forgiveness, and promises not

to offend in that way again, then each of her steps—apology, contrition, signs of remorse and regret (like crying), sincere empathy, offers of restitution, promises renewed fidelity, and humble requests for forgiveness—will reduce the size of the injustice gap (Davis et al. 2016). Perhaps the injustice gap will become small enough to deal with positively.

### 2.3.2. Dealing with Injustice

There are many positive ways to deal with injustices (Wade and Worthington 2003), and some that are not so positive (i.e., revenge) (McCullough 2008). For example, the offended and hurt person could turn the issue over to God expecting God to enact Divine justice on the person's behalf or simply to relinquish it to God. The person could rely on societal or restorative justice (Van Tongeren et al. 2012). The offended and hurt person could forbear, which is to restrain oneself from responding negatively, usually to preserve group harmony (Lin et al. 2019). Forbearance is usually a preferred response in collectivistic cultures. The person could accept that life happens and attempt to move on with life. All of those ways can reduce the size and emotional loading the perceived injustice gap. The injustice gap might become small enough to simply let go of the injustice.

But, if people cannot reduce the injustice sufficiently to relinquish it, or if they keep refreshing the emotional loading through rumination, then they might begin to experience emotional unforgiveness. *Emotional unforgiveness* is feeling a combination of resentment, bitterness, hatred, hostility, anxiety, and anger. The unique combinations of those emotions are perceived within our working memories as unforgiveness (Exline et al. 2003). Once we experience unforgiveness, as Christians, we are called upon to forgive.

### 2.3.3. Definition of Two Types of Forgiveness and of Reconciliation

Historically, experts in various fields have disagreed about how forgiveness is defined (Worthington 2005). This is generally the case whenever a new field falls under scrutiny. Precise definitions for psychologists who study forgiveness are a must because definitions guide future research. Initially, psychologists disagreed, but finally some consensus emerged—at least within psychological research. By 2005, psychologists had come to substantial agreement (Worthington 2005). First, they clearly understood what forgiveness is not. It is not seeing justice done by the courts or natural consequences. Nor is it accepting that "bad stuff happens" and moving on with life. It is not forgetting. If one forgets, forgiveness is not necessary, nor indeed possible. It is not condoning the offense, which is saying that the transgression was in fact the right thing to have done. Forgiving is not justifying what was done, that is, concluding that one had just cause to do wrong. It is not excusing (i.e., deciding that extenuating circumstances were enough that one might overlook the wrongdoing). It is not forbearing—not responding to provocation for the sake of group harmony. Forgiving is not equivalent to reconciling, which is restoring trust in a relationship after trust had been violated. Forgiving is not simply saying, "I forgive you." After all, one can say the words and not mean them and in fact be setting up the offender for revenge.

Second, gradually, it became obvious that there were two types of forgiveness. These are not seen as two halves of a unitary concept of forgiveness but rather as two separate experiences, which are capable of being experienced separately but are often related to each other. *Decisional forgiveness* is making a decision about one's behavioral intentions toward the offender (Davis et al. 2015b; Exline et al. 2003). One decides to treat the person as a valued and valuable person and forgoes revenge. Importantly, decisional forgiveness is not a behavior. It is an intention about how one will behave in the future, and even if the offender were to die and the person was unable to carry out behavioral intentions, decisional forgiveness remains valid.

One might decide to forgive an offender and still be emotionally unforgiving of the person. That is, the victim might be unable to get over the emotional trauma of the offense and still harbor anger, anxiety, resentment, or the other components of emotional unforgiveness. The decisional forgiveness is still valid, but this suggests that there is a second type of forgiveness. *Emotional forgiveness* is defined as the

emotional replacement of negative unforgiving emotions with positive, other-oriented emotions like empathy, sympathy, compassion, or love for the offender (Exline et al. 2003). Emotional replacement is facilitated by several non-self-focused emotions like gratitude for having been forgiven oneself, humility (including recognition that I, too, have erred), and hope (i.e., desire for a better future). Evidence supporting the emotional replacement hypothesis is summarized in Worthington (2006).

Forgiveness in close and continuing relationships is different than forgiveness in stranger, non-continuing, or non-valued interactions (Worthington 2006). In relationships with a past and a future (e.g., marriage- and family-like relationships), forgiveness can be either harder or easier than with relationships that involve either strangers or emotionally distant others. For strangers who hurt us (e.g., someone who robs us) or people we do not want to continue to interact with (e.g., a toxic former boss or a disagreeable and contentious ex-romantic partner), we are generally content to call *complete* emotional forgiveness simply eliminating negative feelings toward the person. However, with someone in a continuing, valued relationship, we are not content with eliminating negativity alone, but we insist that complete emotional forgiveness must restore a net positive valence to the relationship. Thus, we seek not only eliminating negative unforgiveness, but also we want to be empathic, humble, and take responsibility for our own misdeeds within the relationship. This makes forgiving a valued person harder in some ways than forgiving a stranger. In addition, forgiving strangers is easier than forgiving a loved one because the stranger is not around to compound the problem with subsequent interactions. However, in other ways, forgiving strangers is harder than forgiving valued relationship partners. Strangers are not around to reduce the injustice gap through apologies, making amends, requesting forgiveness, or offering forgiveness after one has taken responsibility for one's own offenses. People have little emotional bond toward strangers. Thus, the motivation to forgive, or to keep on forgiving after reducing unforgiveness to zero, is often next to nil.

Decisional and emotional forgiveness are two separate experiences. Either might occur independently of the other. Either might occur first in time and pull the other along. It is easier for most people to see that making a decision to act differently toward an offender might stimulate emotional replacement of negative emotions with more positive emotions. But the reverse order is also possible. For example, if a person held a grudge against an offender, and the offender were hit by a truck breaking all of the offender's bones and throwing the person in chronic pain for the future, then sympathy would likely well up in the grudge-holder, replacing some of the resentment for the minor offense and leading to emotional forgiveness, even though no decision to forgive had yet been reached. Later, the grudge-holder might decide to treat the offender differently, experiencing decisional forgiveness. Although there is no conceptual reason that emotional and decisional forgiveness must be related, many studies have shown that usually a correlation of about 0.4 exists between the two, so in practice the two are conflated.

Often people confuse decisional and emotional forgiveness with reconciliation (Fehr et al. 2010; Freedman 1998). *Reconciliation* is defined as the restoration of trust in a relationship where trust has been violated (Worthington and Drinkard 2000). Whereas forgiveness is something that occurs inside of one's skin, reconciliation occurs between people. For reconciliation to occur, trust must be restored. But if one person refuses or is unable to be trustworthy, then trust appropriately will not and should not occur. So, reconciliation and both types of forgiveness are independent of each other. A person could experience complete decisional and emotional forgiveness, for example, of a deceased parent or an ex-spouse and yet not experience any reconciliation. When people do think (incorrectly) that one must somehow reconcile in order to forgive, that can lead to dangerous consequences. An abused woman who is encouraged (incorrectly) that, in order to fully forgive her partner, she must return and reconcile with him, might place herself in a dangerous position. Instead, she should be encouraged to make a decision to forgive, and she might work to experience emotional forgiveness, but she should not be encouraged to reconcile as long as there is the slightest danger of harm to her from an unrepentant mate.

### 2.3.4. Christian Biblical Theology of Forgiveness: Some Speculative Implications

According to Christian New Testament scriptures, forgiveness is mandated. In Matt 6:12, 14–15, Jesus is teaching his disciples to pray. He prays, "[12] And forgive us our debts, as we also have forgiven our debtors . . . [14] For if you forgive other people when they sin against you, your heavenly Father will also forgive you. [15] But if you do not forgive others their sins, your Father will not forgive your sins."

In this scripture, there are three important theological implications about forgiveness and reconciliation. First, we can see how essential it is to separate reconciliation from forgiveness. If reconciliation were included in the process of forgiveness, then a person who refused to be trustworthy could hold a willing forgiver hostage, preventing forgiving. Also, a woman who is being consistently abused by a spouse would be required to put herself in physical danger in order to forgive. But if a decision to forgive is an event apart from reconciliation, and reconciliation depends on mutual trustworthiness, then the person who is trying to hold one hostage by refusing to reconcile can still be forgiven. And the woman who is being abused can still forgive. But reconciliation—while, in many ways is desirable—might simply not happen if both partners are not trustworthy. Paul argues, "If possible, so far as it depends on you, live peaceably with all" (Rom 12:18). As Paul correctly notes, it is not always possible to restore peace, to reconcile, because it does not depend on individuals alone.

Second, Jesus' commands are not conditional. People are to forgive unconditionally. Some theologians point to Luke 17:3, "Pay attention to yourselves! If your brother (or sister) sins, rebuke him, and if he repents, forgive him." The pertinent question, then, is what if your brother (or sister) does not repent? We are to forgive nevertheless. Human interpersonal forgiveness is unconditional.

Third, we suggest that, in the Christian Scriptures, Jesus is talking about decisional forgiveness, not emotional forgiveness (Worthington 2003). We can see decisional forgiveness in scripture when Peter asks Jesus how often must we forgive, and Peter suggests seven times (Matt 18:21). "Jesus said to him, 'I do not say to you seven times, but seventy-seven times'" (Matt 18:22). That is, every time. It is quite improbable to come to peace in our emotions if we are offended frequently—even a literal 77 times. It would be quite harsh to consider the consequences of Matthew 6:14 and 15 if we had to squelch all shades of negative emotion after every offense. However, if what is required is a decision about our behavioral intentions, that is difficult with repeated offenses, but not near impossible.

### 2.3.5. Forgiveness in Non-Christian Religions

Other religions also have theologies of forgiveness. Dorff (1998) has described a Jewish theology of forgiveness put forward by Maimonides. Essentially, Dorff argues that a Jewish person who is offended can willingly and unilaterally forgive, but theology also mandates forgiveness under certain conditions when the offended party is reluctant to forgive. Forgiveness is typically contextualized within *tsheuvah* (i.e., repentance) of the offender. An offender has left the path of God through an offense and may seek return through a series of steps. These involve apologizing, being contrite, offering or making restitution, demonstrating that one would not offend again given similar circumstances, and asking sincerely for forgiveness. According to Maimonides, forgiveness does not have to be granted unless the offender asks sincerely three times. If forgiveness is refused on the third time, the offended person has left the path of God, and must pursue forgiveness from the other. The necessity of an offender making amends by offering restitution, asking the offended for forgiveness, and receiving a verbal pardon suggests that in Judaism, there are unforgivable offenses, like murder (where the offender cannot make things right with a dead victim and the victim cannot grant forgiveness) or character assassination, which simply cannot ever be made right.

Forgiveness is also valued in Islam, but an offended person is entitled to justice. Forgiveness is thus seen as something extraordinary and blessings befall the forgiver who puts rightful justice aside to pursue forgiveness (Rye et al. 2000). In addition, Rye and his colleagues have summarized philosophies or theologies of forgiveness also from Buddhism and Hinduism, respectively. All major religions value forgiveness, but they understand it differently and practice it differently.

## 2.4. Humility

Earlier, we pointed to humility as a facilitative condition to promote forgiveness. We are not willing to say that humility is required for forgiveness. Many arrogant people sometimes forgive others who have wronged them. But humility certainly helps people forgive. The reason is within the definition of humility. Psychologists have been studying humility since 2000 (Worthington and Allison 2018). Acceptable definitions of humility have gone through a few modifications but currently, Worthington and Allison define humility as having three parts. Humility includes (1) having an accurate self-appraisal, including awareness of one's limitations (including our awareness that there is one God and it is not we); (2) presenting ourselves modestly (not too high nor too low); and (3) being oriented toward others, which is manifest as power under control to lift others up and not squash them down.

Humility might just be the queen of the virtues as Augustine believed. He said, "Humility is the foundation of all the other virtues hence, in the soul in which this virtue does not exist, there cannot be any other virtue except in mere appearance." We can understand his reasoning if we examine a few virtues. For example, courage involves physically laying down one's life for others in spite of fear. Justice involves voluntarily self-limiting one's freedom or privileges to provide equity for all. Forgiving and mercy involve giving up one's legitimate right to justice. Altruism involves acting with unmerited favor at cost to oneself. Compassion involves pouring one's life and resources out to help someone who needs it. Wisdom involves cultivating discipline to gain perspective to help others. The common denominator seems to be this: In difficult situations, recognizing accurately one's position, acting modestly, and then pouring oneself out in other-oriented service usually in times of testing. Furthermore, Davis et al. () showed that humility is crucial in repairing damaged social and emotional bonds, which is vital in couple relationships (Davis et al.).

## 2.5. Forgiveness and Relational Spirituality

Forgiveness has a spiritual level to it that should not be neglected—especially in religious people (Davis et al. 2008). At the strictly interpersonal level, forgiveness depends on the relationships between victim and offender, victim and offense, and offender and offense. That is, the ease with which one forgives (or does not forgive) depends on, for example, the closeness and quality of the relationship between victim and offender. But just as much, the ease with which one forgives (or does not) depends on the relationship between the victim and the offense (e.g., was the offense harmful, are reminders of it present?) and between offender and the offense (e.g., is this a frequently repeated offense or a one-off offense?).

At the spiritual level, there is a relationship between God and the victim. Does the victim have an attachment to God? Is the person in a loving and dependent relationship to God? Does the victim have a relationship that is stable (i.e., dwelling with God), or is the person unstably connected to God or perhaps is a habitual seeker (Wuthnow (1998); for a review of research on forgiveness and relational spirituality, see (Worthington and Sandage 2016))? However, the offender also has a relationship to God and the victim assesses it: Does the offender have a similar or different relationship to God as I do (Davis et al. 2008)? Worthington and his colleagues have found, for example, that Christians who are offended by another Christian believer experience more sense of harm from the same transgression than if they had been offended by a non-Christian (Greer et al. 2014). If a Christian bilks another Christian out of $10,000, it would hurt more than if a non-Christian did the same crime. On the other hand, people are at the same time more willing to forgive a like-minded believer than one who is thought to be religiously and spiritually dissimilar (Greer et al. 2014). People do not just evaluate the similarity of religious identification (e.g., we are both Christians), but they also evaluate the similarity in religious beliefs, values, and practices.

There is a sacred dimension to the offense (Pargament et al. 2005). Some offenses are seen as desecration of something sacred or a sacred loss, and those are particularly hard to forgive. For example, if a partner considers the marriage to be a sacred bond, and their spouse has an affair, then it is imbued

with more hurtfulness because the spouse desecrated the sacred marriage bond. It compounds the hurt if the person then shatters the bond bringing about a sacred loss of the marriage bond in addition to the desecration of the marriage bond.

Thus, when one is evaluating whether forgiveness is to be extended to an offender, one considers the relationships—both sacred and secular—among victim, offender, transgression, and those things people hold sacred. Worthington and Sandage (2016) have outlined the theorizing and counseling implications for relational spirituality and forgiveness. Their theorizing has been based on a research program of over 40 scientific studies. These studies include studies of the REACH Forgiveness model.

*2.6. Five Steps to Reach Emotional Forgiveness of the Partner*

Over the last 25 years, Worthington (2003, 2006) and his colleagues have been investigating the REACH Forgiveness method of helping people forgive. These investigations have occurred in psychoeducational groups, church groups, groups in college dormitories, groups of partners, helpers meeting conjointly with partners for enrichment, couple and individual psychotherapists meeting with their clients, and even do-it-yourself workbooks. The resources to conduct these groups are available without cost (see www.EvWorthington-forgiveness.com). Note that, as we discussed above, forgiveness is independent of reconciliation. Furthermore, one might experience a decision to forgive without substantial emotional forgiveness, and sometimes one might experience emotional forgiveness without ever having made a decision to forgive.

2.6.1. Description of the Steps of REACH Forgiveness

The REACH Forgiveness method involves several steps, which we will outline as if a forgiveness intervention were being done in a psychoeducational group of about six hours duration. Small modifications in order and spontaneity are needed in individual or couple counseling, in psychotherapy process groups, or in only loosely led small group settings (Wuthnow 2000). First, to create some sense of efficacy, the leader asks participants to describe the most difficult offense they have ever successfully forgiven. Second, participants are helped to define forgiveness as two experiences—decisional and emotional forgiveness. Third, participants reflect on the benefits of forgiving—benefits to the self and to the offender (i.e., physical, mental health, relational, spiritual, and moral). Fourth, participants are invited to make a decision to forgive. Leaders acknowledge that this might be too early for many to make such a decision, and group members are told that this will be reconsidered near the end of the group. Fifth, participants are led through five steps to REACH Emotional Forgiveness. These take the most time (about four to five of the six hours). The steps are as follows. R = Recall the hurt but in a way that does not rehearse grudge-holding or victimization. E = Empathize with the offender, or more generically emotionally replace negative emotions through empathy, sympathy, compassion, or love. A = give an Altruistic gift of emotional forgiveness to the person who hurt or offended. C = Commit to the forgiveness experienced through writing a contract to oneself or doing some ritualistic behavior (e.g., symbolically washing a description of the harm from one's hand or writing a brief description and pinning it to a cross). People then make a public commitment so that they will be able to H = Hold onto the emotional forgiveness they experienced even when they doubt that they have in fact forgiven. Sixth, people are invited to reconsider whether they wish to make a decision to forgive, if they have not done so already. Seventh, and finally, people are led through 12 exercises that help them generalize their use of the REACH Forgiveness method to other offenses and harms.

2.6.2. People Can Be Helped to Forgive

Numerous studies have been done in testing the efficacy of the REACH Forgiveness model when compared to either a no treatment waiting list or an alternative treatment. A meta-analysis of intervention studies compared all outcome studies that randomly assigned people to treatment (or none or alternatives; Wade et al. (2014)). Randomized controlled trials (RCTs) are considered the gold standard of intervention research. A meta-analysis is a scientific study that reduces all qualifying

studies to a single metric (i.e., number of standard deviations of change or an effect size, called *d*). There were 67 outcome studies found: for REACH Forgiveness (*n* = 22); for Enright's Process (*n* = 23); for all others (*n* = 22). Of those, 52 were RCTs qualifying for analysis: REACH Forgiveness (*n* = 18); Process (*n* = 20); all others (*n* = 14). The major finding was that the effect size was about *d* = 0.1 per hour of intervention. That is, efficacy of treatment was proportional to time in treatment. It did not matter whose forgiveness intervention was considered. It mattered only a little how the treatments were delivered. Individuals in psychological treatment did a little better than those who attended group treatment. Individuals did not differ from couples seen by a counselor.

### 2.6.3. Forgiveness Groups Stimulate Other Mental Health Benefits

Wade et al. (2014) considered the effects of participating in a forgiveness treatment on outcomes besides forgiveness—even though the other outcomes were not the intention of the forgiveness treatments. In ten studies that measured depression, the effect size for depression was *d* = 0.34 and the effect size of forgiveness was *d* = 0.60. In seven studies that measured anxiety, the effect size for anxiety was *d* = 0.63, and for forgiveness it was *d* = 1.34. In six studies that measured hope, the effect size for hope was *d* = 1.00, and for forgiveness *d* = 0.94. Thus, in forgiveness studies, the effect size for depression and anxiety is about half the effect size for forgiveness, and the effect size for hope is about the same as for forgiveness.

When people participate in forgiveness interventions, their depression and anxiety get better. If we put this in context, the result seems more important. If a person attends about 26 weeks of focused cognitive behavior therapy, the mean effect size is about *d* = 1.24 when targeted at reducing depression (compared to *d* = 0.34 for participating in a 6-h forgiveness group) and about *d* = 0.9 when targeted at reducing anxiety (compared to *d* = 0.63 for participating in a six-hour forgiveness group)! By working to forgive, the forgiveness results in more than half the gains in either direct treatment for depression or anxiety, even though the forgiveness treatments do not directly try to change depression or anxiety.

Let us be clear what we are claiming (as well as what we are not claiming). We are not claiming that forgiveness is causal for decreasing depression and anxiety. Rather, we are claiming that, in a meta-analysis that examined different intervention programs to promote forgiveness, that intervention program (regardless of which clinical scientist developed it) also produced decreased depression and anxiety.

### 2.6.4. Do-It-Yourself Workbooks

Workbooks that take six to seven hours to complete can produce gains in forgiveness equal to groups. Recall, Wade et al. (2014) found *d*~0.1 per hour of treatment for group, individual, or couple face-to-face treatment. Recently Worthington and his colleagues have completed three studies that demonstrate comparable results using individually completed workbooks. Two have involved secular students. In Harper et al. (2014), there was a *d* = 0.08/h change for forgiving an identified hurt. In Lavelock et al. (2017), a *d* = 0.08/h change for trait forgiveness, not a specific instance of forgiveness. In a Christian workbook study by Greer et al. (2014), for Christians who have been hurt by other Christians, there was a *d* = 0.2/h change, which is twice the effect in secular treatments. We must tentatively conclude from this very preliminary evidence that do-it-yourself workbooks might even be stronger interventions to promote forgiveness than are live treatments.

### 2.7. Self-Condemnation and Self-Forgiveness

### 2.7.1. Self-Condemnation in Close Relationships Is Commonplace and Harmful to Relationships

When victims or God extend forgiveness to offenders it bridges the injustice gap. Often offenders feel emotional relief, and reconciliation is promoted. Yet, people often chronically hold on to self-condemnation, and this is especially true when offenders have harmed people most close to

them, such as a romantic partner. In Griffin et al. (2015a), students identified things that they had done in that past that they still felt guilty about. About 80% of people had hurt a family member, romantic partner, or close friend. Most of offenses people recalled had occurred more than one year prior to the data collection (Griffin et al. 2015a). When people struggle with self-condemnation, it can lead to continued offenses in close relationships and worse off relationship satisfaction (McConnell 2015; for reviews see Woodyatt et al. 2017).

### 2.7.2. We Can Experience Self-Condemnation Even after Close Others or God Forgives Us

Although the injustice gap is narrowed with forgiveness, people still struggle with self-condemnation because feeling forgiven is often not enough to remove the experience of guilt and shame. There is both an interpersonal and intrapersonal process to self-forgiveness. Although self-forgiveness is promoted with interpersonal restoration, the internal psychological process of developing self-benevolent beliefs, feelings, and actions is at the core of self-forgiveness (McConnell 2015). To fully engage in these dual routes to self-forgiveness, people need humility. It takes humility to admit faults and limitations and to commit to continually working on oneself, and to do so in a manner that adheres to justice, respect of others, and prosocial motivations rather than self-serving motivations. People often struggle with this in-depth process due to its inherently humbling nature. While Christians may intellectually understand they are forgiven by others and God, they may not emotionally internalize the benevolence extended to them. McConnell and Dixon (2012) found a stronger relation between perceived forgiveness from God and self-forgiveness when people believed God could forgive them personally, but the relation was weaker when they had a broad theological conviction that God forgives people in general. This suggests that Christians who are solid in their understanding of forgiveness-related scripture may not emotionally experience God's forgiveness. Even so, experiencing forgiveness from God personally did not fully, or even near fully, explain the experience of self-forgiveness. Multiple studies also have found that self-condemnation persists on despite feeling forgiveness from victims in secular contexts (McConnell 2015).

Like psychotherapy and psychopharmacological treatments, self-forgiveness is not a Biblical concept. The Bible focuses on God's forgiveness of our moral guilt. But, just because the Bible does not mention it does not mean it is not psychologically important—just as the Bible does not name post-traumatic stress disorder, borderline personality disorder, resilience, and a host of psychological constructs that are clearly within psychological experience and important to address in the spiritual life of parishioners. Self-forgiveness, while not a Biblical concept, is helpful in spiritual sanctification, behavior modifications, and overcoming religious and spiritual struggles (McConnell 2015; Woodyatt et al. 2017).

For example, King David, of course, messed up royally. After being confronted by Nathan the prophet, David confessed to Nathan and Nathan immediately conveyed God's forgiveness of David. In 2 Sam 12:13, we read, "David said to Nathan, 'I have sinned against the LORD.' And Nathan said to David, 'The LORD also has put away your sin; you shall not die.'" Yet, later, we find in the note at the beginning of Psalm 51, that famous song of lament, *"To the choirmaster. A Psalm of David, when Nathan the prophet went to him, after he had gone in to Bathsheba."* We are familiar with the self-condemnation David expressed in that Psalm,

> 1 Have mercy on me, O God,
> according to your steadfast love;
> according to your abundant mercy
> blot out my transgressions.
> 2 Wash me thoroughly from my iniquity,
> and cleanse me from my sin!
>
> 3 For I know my transgressions,
> and my sin is ever before me.

4 Against you, you only, have I sinned
and done what is evil in your sight,
so that you may be justified in your words
and blameless in your judgment.
5 Behold, I was brought forth in iniquity,
and in sin did my mother conceive me.
6 Behold, you delight in truth in the inward being,
and you teach me wisdom in the secret heart.
7 Purge me with hyssop, and I shall be clean;
wash me, and I shall be whiter than snow.
8 Let me hear joy and gladness;
let the bones that you have broken rejoice.
9 Hide your face from my sins, and blot out all my iniquities.
10 Create in me a clean heart, O God,
and renew a right spirit within me.
11 Cast me not away from your presence,
and take not your Holy Spirit from me.
12 Restore to me the joy of your salvation,
and uphold me with a willing spirit.

David had been forgiven his sins by God, and Nathan, speaking directly on behalf of God, had confirmed it. Yet, David apparently still felt self-condemnation much later. And it is likely there is good reason for that self-condemnation. The social and psychological effects of David's sin were not removed by God's forgiveness of David. Those effects dogged him for the rest of his life, and frankly, forgiven by God or not, he suffered from them. David knew of God's forgiveness, yet he struggled with self-condemnation.

### 2.7.3. A Two-Factor Theory within the Dual Process Model of Self-Forgiveness

Sometimes people perceive self-forgiveness as egocentric because they confuse self-forgiveness with denial, justification, self-exoneration, or easily letting oneself off the hook. Actually, responsible self-forgiveness is when people replace self-condemnation with self-benevolent beliefs, feelings, and actions toward themselves while reaffirming values, seeking amends, and making genuine behavior changes (McConnell 2015). Griffin et al. (2015a) advanced a two-factor theory of self-forgiveness. They suggested that self-forgiveness is composed of moral reaffirmation of values and restoration of positive self-regard. Reaffirming one's values without restoring positive self-regard leaves one mired in self-condemnation. Restoring positive self-regard without reaffirming one's values is irresponsibly letting oneself off of the hook. Both aspects of self-forgiveness are needed.

This two-factor theory maps well onto McConnell (2015)'s dual process of interpersonal and intrapersonal restoration of self-forgiveness. Reaffirming one's values helps both offenders and victims know of responsibility taking, sincere intentions (e.g., forbearances, behavioral intentions), and guilt in the process of conciliatory behaviors and overall interpersonal restoration. Reaffirmation of values also in and of itself helps offenders remember they are capable of doing good and made in the image of God, thus promoting positive self-concepts in the process of intrapersonal restoration. Restoring positive self-regard characteristically leads to behavior change in the process of intrapersonal restoration. Initial and continued behavior change leads to greater chances that victims will express forgiveness and promote interpersonal restoration (McConnell 2015).

The moral imperative for self-forgiveness and its importance in close relationships is highlighted by the effects of responsible self-forgiveness. McConnell (2015) clarified that responsible self-forgiveness can lead to decreased problematic behaviors in the future. McConnell (2015) discussed how chronic self-condemnation and intropunitive behaviors can actually unintentionally lead to

increases, rather than decreases, in problematic behaviors. In fact, some preliminary evidence has found that responsible self-forgiveness—not easily letting oneself off the hook—reduces the likelihood of repeat offenses as well as increases prosocial behaviors and improved relationship satisfaction. When self-forgiveness is coupled with value reaffirmation and motivation for change, it can lead to better physical, emotional, spiritual, and relational outcomes (McConnell 2015; Woodyatt et al. 2017).

Responsible self-forgiveness involves both moral affirmation and restoring positive self-regard at both the interpersonal and intrapersonal levels. The process of self-forgiveness requires humility, justice, and an attempt at reconciliation in close relationships. Moral reaffirmation of values involves humbly going to victims and God with one's sin and receiving mercy and grace, seeking to repair social damage, and resolving to live virtuously (with God's help) in the future to promote harmonious relationships. Restoring positive self-regard involves reducing negative rumination, REACHing emotional self-forgiveness (using the REACH Forgiveness model applied to oneself), realizing self-acceptance through a restored concept of oneself despite one's specific wrongdoing, and more generally self-acceptance of one's fallen nature and capacity for doing wrong with a commitment to behavior change.

### 2.7.4. Putting the Two-Factor and Dual Process Model into a Helping Framework: Six Steps to Self-Forgiveness

Worthington (2013) actually created a helping model to promote responsible self-forgiveness prior to Griffin, Worthington, Lavelock et al. and McConnell articulating their respective processes of self-forgiveness. The first three steps, Worthington grouped under a heading of Responsibility. Step 1 was to Receive God's Forgiveness. Step 2, aimed at repairing social damage, was entitled Repent and Repair Relationships. In Step 3, the psychological damage was addressed. Although it was named Reduce Rumination, it also sought to promote examining unrealistic expectations and reducing rumination. At the middle of the model was Step 4, REACH Emotional Self-forgiveness. The person was coached to apply the five steps in the previously described REACH Forgiveness method but aimed toward themselves. In the third and final part of the six steps, two steps comprise Repair of Self. Step 5 is to Realize Self-Acceptance. Step 6 is to Resolve to Live Virtuously.

Griffin et al. (2015a) tested the efficacy of this model of self-forgiveness in a six- to seven-hour do-it-yourself workbook (www.EvWorthington-Forgiveness.com) by using a wait-list control design. Most people made substantial gains in both reaffirming their moral values and restoring their positive self-regard. Essentially, people in the immediate treatment condition forgave themselves from the initial test to the post-workbook test, and they maintained their gains for at least two weeks. People in the delayed treatment condition did not change during the waiting period, but subsequently changed similarly to the immediate treatment condition after eventual completion of the workbook. Whereas the workbook is not tailored to Christians, much research on religiously accommodated treatments shows an interesting and relative finding for treatment considerations (see Captari et al. 2018). People who are Christians do equally well at changing their mental health with secular or explicitly Christian treatments but do better at changing their spiritual well-being when they are in the explicitly Christian-accommodated treatments.

### 2.8. Why Promote Forgiveness in Long-Term Couple Relationships

In this final section, we argue that forgiveness and self-forgiveness are strongly related to health directly due to reduced stress. A stress-and-coping theory of forgiveness suggests that transgressions are stressors that are perceived to be harmful. People respond physiologically, emotionally, cognitively, and behaviorally to the appraisal of threat, and that triggers coping. Stress has been shown to have strong effects on physical health. It does so by increasing peripheral nervous system activation (e.g., blood pressure, heart rate, skin conductance), cortisol (and high levels consistently affect virtually every bodily system; (Sapolsky 2004)), lowers heart rate variability (which affects parasympathetic nervous system's calming functions), and central nervous system functioning (affecting memory,

amygdala responsiveness, etc.). Coping, in turn, affects physical health in the long run (for reviews, see Toussaint et al. 2015).

However, in addition to directly affecting physical health, forgiveness and self-forgiveness helps improve mental health, and that has been shown to be related to physical health. Griffin et al. (2015b) reviewed the positive relation between forgiveness and mental health, and the indirect impact forgiveness has on physical health by reducing rumination and thus risk for depression, anxiety, anger-related disorders, obsessive-compulsive disorders, post-traumatic stress disorders, and even psychophysiological disorders. When Davis et al. (2015a) reviewed the existing correlational research, they found self-forgiveness related to both physical and psychological well-being.

Also, forgiveness and self-forgiveness has been related to better couple relationships, and marriage and marriage-like relationships has been shown to be strongly related to health (Waite and Gallagher 2001). Better relationships not only provide social support of various kinds, but they reduce stresses by resulting in shared activities, intimacy, reduced economic pressures, and shared work.

Finally, unforgiveness can put people out of harmony with whatever they hold to be sacred. Forgiveness helps restore people to spiritual harmony. As we mentioned, forgiveness is valued by all major religions (Rye et al. 2000). When religious people are unable or unwilling to forgive, that creates a disruption of spiritual harmony. Thus, forgiveness increases religious and spiritual engagement. McCullough et al. (2000) meta-analyzed religion and longevity. They showed that even with seven potential confounds controlled, religion was still related to longevity as strongly as stopping smoking a pack and a half of cigarettes a day was related to longevity. Therefore, religiously oriented help-providers can and should promote better relationships and health through promoting forgiveness.

## 3. Discussion

Transgressions—whether others inflict them on us or we inflict them others—create pain and suffering. There is nothing noble about pain and suffering per se, but the way we deal with them can redeem the pain and suffering by forming our character. As Brooks (2015) says in *The Road to Character*, "Most people shoot for happiness but feel formed through suffering" (p. 93).

Suffering pushes us into new unexplored dark spaces. If we explore that dark space instead of letting events just carry us with the tornado-like destructive winds, the suffering can have some good outcomes along with inevitable destructive effects. For example, suffering can help us understand ourselves, our limitations, our ultimate lack of control, and our dependence on God. It can help us see our solidarity with others who suffered through the ages. It can help us experience a sense of gratitude for the help of others. It can call us—if we but hear—to respond morally, spiritually, meaningfully, and with holiness to the suffering. Notice, the redemption of transgressions and suffering depend on humility before God and before others. As we said earlier, humility consists of three parts—an accurate self-appraisal including our place before God and our sense of limitations, a modest self-presentation, and an orientation to build others up.

Humility is vital. We do not heal unscarred from transgressions through forgiveness. We emerge different. Forgiveness is not forgetting. It is remembering differently. Humility helps us direct forgiveness toward building others up, giving an altruistic gift of forgiving. Humility helps us face our own sins and deal with them effectively through responsible self-forgiveness. Transgressions are potentially devastating, but they can be a "dreadful gift"—that forms more clearly in us Christ-likeness if we are able to give the altruistic gift of forgiveness to others and experience responsible self-forgiveness with justice. Forgiving is for giving, not for getting.

We have tackled a few important themes in this article. As we saw from the Good Samaritan parable, theologians, clergy, and psychological help-givers can work together (or certainly one after another) to promote forgiveness. We can help offenders reduce the injustice gap by helping them see the helpfulness of apologies and amends making. We can help those transgressed against to forgive by using the REACH Forgiveness method (and other methods, too). When people offend and cannot

turn loose of self-condemnation, we can help people forgive themselves responsibly. Forgiveness is one main way that couples can restore damaged emotional bonds—starting from the side of both transgressor and transgressed. Through forgiveness that is mindful of justice, couples can improve each other and reconcile in love. Let us suggest that love is woven together with humility and forgiveness into a three-fold cord that is not quickly nor easily broken (Eccl. 4:12). That cord is intertwined love, humility, and forgiveness.

**Author Contributions:** Conceptualization, all three co-authors contributed most heavily within their specialty areas: E.L.W., forgiveness, couples, and couple therapy; E.M.B., theology, pastoral counseling, and relationships of theologians, pastors, and psychological help givers; J.M.M., self-forgiveness, psychotherapy, and forgiveness. Writing—Original Draft Preparation, E.L.W.; Writing—Review & Editing, E.M.B. and J.M.M.

**Funding:** This research received no external funding.

**Acknowledgments:** The original manuscript was prepared by expanding and refocusing the script of a keynote address by E.L.W. to the Evangelical Theological Society, 2015.

**Conflicts of Interest:** The authors declare no conflict of interest.

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
