# Peer review of "Forgiveness in Committed Couples: Its Synergy with Humility, Justice, and Reconciliation"

_religions, doi:10.3390/rel10010013_

Round 1
Reviewer 1 Report
This article rightly calls for pastors and counselors to work more closely together and gives an example of one area in which collaborative and cooperative work can be extremely fruitful: forgiveness in couples counseling. The article carefully and accurately defines forgiveness, identifying its decisional and emotional dimensions and distinguishing forgiveness from reconciliation. Also, the article examines various aspects of the forgiveness experience, including the relative difficulty of forgiving loved ones compared to forgiving strangers, and it explores the connection to the character trait of humility and the feeling of being the victim of an injustice. Moreover, the article includes a brief but helpful description of what Judaism, Christianity, and Islam say about forgiveness and includes an insightful discussion of how self-forgiveness is an important part of the healing and growth process. To illustrate self-forgiveness in a religious context, the author(s) uses the feelings of guilt recorded in Psalm 51, when David is feeling remorse for his sin with Bathsheba. The conclusion includes a discussion of Everett Worthington's theories and materials and the effectiveness of his REACH model of forgiveness, and the author(s) suggests that while the non-religious materials of Worthington and others work well with religious people, materials designed from a faith-based perspective would be even more effective for people who come from religious backgrounds. This article represents only the first steps toward more collaboration between psychology and theology, but it makes a strong case. It will be up to other professionals to fill out the details, some of whom have already been working on this.
Author Response
Thank you for your excellent summary of our main points. It appears that the only comment that we need to respond to is to spell check in some cases that we might have let some spelling errors or typos slip through. We will be sure to do this.
Reviewer 2 Report
These are some issues for the author to consider and elaborate if they feel it will benefit their paper.
1)- The authors make an important distinction among decisional forgiveness, emotional forgiveness and reconciliation. What exactly is the relationship among these? It seems that emotional forgiveness depends on decisional forgiveness. But, perhaps this is not the case. Perhaps someone can emotionally forgive first then arrive at decisional forgiveness. Or, perhaps one can arrive at emotional forgiveness without decisional forgiveness. Does a partner in a committed relationship need to achieve both decisional and emotional forgiveness for there to be reconciliation?
2)- The article is about committed couples (as the title suggests) however, there is no mention of reconciliation in section 2.6 FIVE STEPS TO REACH EMOTIONAL FORGIVENESS OF THE PARTNER. I
3)-In section 2.6.3) the authors make two conclusions from the studies: (1) "When people participate in forgiveness interventions, their depression and anxiety get better"; and (2) "By working to forgive, the forgiveness results in more than half the gains in either direct treatment for depression or anxiety, even though the forgiveness treatments do not directly try to change depression or anxiety." I have two issues with this sections that I hope the authors could modify. First, it is difficult to understand how these conclusions are inferred from the stated data. Could the authors explain it instead of assuming that the readers will easily draw the same inferences they do. Second, I am very suspicious of that such tests actually proving or demonstrating what they claim to. The authors seem to suggest that there is some connection between forgiveness and anxiety and depression. I am worried that the authors are committing a false cause fallacy as in the case of Gotten ratio. If anything, their conclusion "that treatment of forgiveness also helps in anxiety and depression" might be one of various plausible explanations of the data. Are there other plausible explanations? For instance, it might be that people who have issues with forgiveness also have more anger issues as well, and thus as one learns to forgive one's anger subsides which reduces anxiety and depression.
Minor typos
Line 300. “you” should be “your”
493 space between “ispromoted”.
Author Response
I have attached point by point responses to Reviewer 2 as a file.
